# TLR2 mediates renal apoptosis in neonatal mice subjected experimentally to obstructive nephropathy

**Maja Wyczanska**[1], **Jana Rohling**[1], **Ursula Keller**[1], **Marcus R. Benz**[2],
**Carsten Kirschning**[3], **Bärbel Lange-Sperandio**[1]*

**1** Department of Pediatrics, Dr. v. Hauner Children's Hospital, University Hospital, LMU Munich, Munich, Germany, **2** Pediatric Nephrology Dachau, Dachau, Germany, **3** Institute of Medical Microbiology, University of Essen, Essen, Germany

☯ These authors contributed equally to this work.
* baerbel.lange-sperandio@med.uni-muenchen.de

**Data Availability Statement:** The data from this study is available in a public repository (Zenodo) under this DOI: 10.5281/zenodo.8060419.

## Abstract

Urinary tract obstruction during renal development leads to inflammation, tubular apoptosis, and interstitial fibrosis. Toll like receptors (TLRs) expressed on leukocytes, myofibroblasts and renal cells play a central role in acute inflammation. TLR2 is activated by endogenous danger signals in the kidney; its contribution to renal injury in early life is still a controversial topic. We analyzed TLR2 for a potential role in the neonatal mouse model of congenital obstructive nephropathy. Inborn obstructive nephropathies are a leading cause of end-stage kidney disease in children. Thus, newborn *Tlr2*−/− and wild type (WT) C57BL/6 mice were subjected to complete unilateral ureteral obstruction (UUO) or sham-operation on the 2nd day of life. The neonatal kidneys were harvested and analyzed at days 7 and 14 of life. Relative expression levels of TLR2, caspase-8, Bcl-2, Bax, GSDMD, GSDME, HMGB1, TNF, galectin-3, α-SMA, MMP-2, and TGF-β proteins were quantified semi-quantitatively by immunoblot analyses. Tubular apoptosis, proliferation, macrophage- and T-cell infiltration, tubular atrophy, and interstitial fibrosis were analyzed immunohistochemically. Neonatal *Tlr2*−/− mice kidneys exhibited less tubular and interstitial apoptosis as compared to those of WT C57BL/6 mice after UUO. UUO induced neonatally did trigger pyroptosis in kidneys, however to similar degrees in *Tlr2*−/− and WT mice. Also, tubular atrophy, interstitial fibrosis, tubular proliferation, as well as macrophage and T-cell infiltration were unremarkable. We conclude that while TLR2 mediates apoptosis in the kidneys of neonatal mice subjected to UUO, leukocyte recruitment, interstitial fibrosis, and consequent neonatal obstructive nephropathy might lack a TLR2 involvement.

## Introduction

Congenital obstructive nephropathy is a frequent cause of chronic kidney disease in infants and children [1, 2]. Inborn obstruction of the urinary tract impairs renal growth and development and leads to reduced nephron numbers. The reduction of nephrons corresponds with a

**Funding:** The author(s) received specific funding for this work. Deutsche Forschungsgemeinschaft (DFG) 1257/5-1, Prof. Dr. Bärbel Lange-Sperandio.

**Competing interests:** The authors have declared that no competing interests exist.

lifelong risk of end stage kidney disease [3]. Unilateral ureteral obstruction (UUO) in neonatal mice at the second day of life serves as a model for congenital obstructive nephropathy. It studies the effects of urinary tract obstruction on renal development, as nephrogenesis in mice finishes postnatally 2–3 weeks after birth [4]. Contrary, in humans nephrogenesis finishes in utero at 34–36 weeks of gestation. Neonatal UUO elicits tubular apoptosis, renal inflammation, and interstitial fibrosis, which contribute to a loss of nephron mass in the kidney [5]. Inflammatory macrophages, which produce pro-inflammatory cytokines like tumor necrosis factor (TNF), are key players in this process [6]. Toll like receptors (TLRs) are a family of innate pattern recognition receptors. E.g. TLR2 and TLR4 are expressed on leukocytes, myofibroblasts and renal cells which often play a central role in acute inflammation as major sources of pro-inflammatory chemokines and cytokines [7–9]. Besides being implicated as cellular pathogen-associated molecular pattern sensors, TLR might also bind danger-associated molecular patterns (DAMPs) released upon sterile damage of tissue and thus being of endogenous origin [10]. Cognate ligand activated TLRs initiate intracellular signaling cascades such as through myeloid differentiation primary response gene (MyD88)-dependent phosphorylation of MAPK towards activation of nuclear factors such as activating protein-1 (AP-1) and NF-κB [11]. TLRs have been implicated in various renal diseases, including ischemia-reperfusion injury (IRI), wherein endogenous TLR2 and TLR4 ligands are thought to be released from the renal epithelium [8, 12]. For instance, adult $Tlr2^{-/-}$ mice displayed in the IRI model ameliorated kidney inflammation and injury [13]. TLR2 also influenced renal fibrosis, a hallmark of UUO [14].

TLR2 forms heterodimers with TLR1 or TLR6, as well as a variety of further receptors for recognition of diverse ligands [15]. It plays a central role in the innate immune signaling in renal disease [8, 16–18]. During a bacterial infection, TLR2 signals for apoptosis through MyD88 via a pathway involving caspase-8 [19]. High mobility group box 1 (HMGB1) might carry DAMPs that might activate TLR2 [20]. HMGB1 is mostly associated with TLR4 [20, 21]. However, the release of HMGB1 into extracellular fluid also initiates immune responses through TLR2 [22]. Various reports implicate HMGB1 as an important role holder in the pathogenesis of kidney diseases by affecting renal epithelial cell apoptosis, kidney tissue fibrosis, and inflammation [22]. HMGB1 can also trigger pyroptosis, a regulated necrotic cell death, which involves inflammasome activities [23]. Inflammasomes can be activated by DAMPs towards cleavage of gasdermin (GSDM) D or E and consequent cell rupture and release of pro-inflammatory alarmins [23, 24]. The role of TLR2 in the UUO model, inflammatory cell death and fibrosis is being discussed controversially. Renal function of adult $Tlr2^{-/-}$ mice is enhanced as compared to WT controls while $T_H2$ cytokine production and renal fibrosis following UUO are reduced [25]. Here, we comparatively analyzed $Tlr2^{-/-}$ mice for the first time in a neonatal mouse model of congenital obstructive nephropathy.

## Materials and methods

### Experimental protocol

The Tlr2$^{-/-}$ mouse strain used (and crossed with other Tlr ko strains) has been generated by Deltagen, Cal, USA, and provided to CK through Tularik (merged into Amgen in the aftermath) [26]. $Tlr2^{-/-}$ mice and WT mice (C57BL/6) were subjected to complete left ureteral obstruction or sham operation under general anesthesia with isoflurane (3–5% v/v) and oxygen (0.8 L/min) on the second day of life, as described before [27]. The animals received carprofen to alleviate possible pain after the surgery. The sex distribution was equal in both groups. After recovery, neonatal mice were returned to their mothers until sacrifice on day 7 and 14 of life. The animals were sacrificed by cervical dislocation. All experiments were

performed according to national animal protection laws and the guidelines of animal experi-mentation established and approved by the Regierungspräsidium von Oberbayern (Az 55.2-1-54-2531-136-06).

## Identification of infiltrating macrophages and T-lymphocytes

The abundance of infiltrating macrophages and T-lymphocytes in the neonatal kidney was examined by immunohistochemistry. Formalin-fixed, paraffin-embedded kidney sections were subjected to antigen retrieval and incubated with either rat anti-mouse MAC-2 (galectin-3) antibody against macrophages (Cedarlane Laboratories, Canada, CL8942AP, 1:500) or anti-CD3 antibody against T-lymphocytes (Bio-Rad AbD Serotec GmbH, Germany, MCA1477, 1:50). Specificity was assessed through simultaneous staining of control sections with an unspecific, species-controlled primary antibody. Biotinylated horse anti-mouse IgG (Vector Laboratories, CA) was used as secondary antibody. Sections were incubated with ABC reagent, detected with DAB (Vectastain, Vector Laboratories, CA) and counterstained with methylene blue or hematoxylin. Images were taken using the LEICA DM1000 microscope and the digital camera (LEICA ICC50HD, Germany). Macrophages and CD3-positive lymphocytes in cortex and medulla were counted in twenty non-overlapping high-power fields at x400 magnification and were analyzed in a blinded manner (n = 8 in each group). Data were expressed as the mean score ± SEM per 20 high-power fields.

## Detection of apoptosis and proliferation

Apoptotic cells were detected by the terminal deoxynucleotidyl transferase (TdT)- mediated dUTP-biotin nick end labeling (TUNEL) assay, as described before [5]. Briefly, 4% formalin-fixed tissue sections were deparaffinized and rehydrated in ethanol, followed by incubation with proteinase K. After quenching, equilibration buffer and working strength enzyme (Apop-Tag Peroxidase In Situ Apoptosis Detection Kit, Millipore, MA) were applied. Cells were regarded as TUNEL-positive if their nuclei were stained black and displayed typical apoptotic morphology. Apoptosis in each kidney was calculated by counting the number of TUNEL-pos-itive tubular and interstitial cells in 20 sequentially selected fields at x400 magnification in a blinded fashion and expressed as the mean number ± SEM per 20 high-power fields using the LEICA DM1000 microscope and the digital camera (LEICA ICC50HD, Germany). For detec-tion of proliferation formalin-fixed, paraffin-embedded kidney sections were subjected to anti-gen retrieval and incubated with mouse anti-rat Ki67 antibody (Dako, # M7248, Agilent Technologies, CA) at 1:50. Sections were incubated with ABC reagent, detected with DAB (Vectastain, Vector Laboratories, CA) and counterstained with hematoxylin. Digital images of the sections (n = 8 in each group) were superimposed on a grid, and the number of dark brown Ki67 positive nuclei was recorded for each field. Proliferating tubular and interstitial cells in cortex and medulla were counted in twenty non-overlapping high-power fields at x400 magnification and were analyzed in a blinded manner using the LEICA DM1000 microscope and the digital camera (LEICA ICC50HD, Germany). Data were expressed as the mean score + SEM per 20 high-power fields.

## Measurement of tubular atrophy

Kidney sections were stained with periodic acid Schiff for assessment of tubular basement membranes, and tubular atrophy was determined as described previously [5]. Atrophic tubules were identified by their thickened and sometimes duplicated or wrinkled basement mem-branes. Digital images of the sections (n = 8 in each group) were superimposed on a grid, and the number of atrophic tubules was recorded for each field. Twenty non-overlapping high-

power fields at x400 magnification were analyzed in a blinded fashion. Data were expressed as the mean score ± SEM per 20 high power fields.

## Measurement of interstitial fibrosis

Interstitial collagen deposition was measured in Masson's trichrome-stained sections. Digital images of the sections were superimposed on a grid, and the number of grid points overlapping interstitial blue-staining collagen was recorded for each field in a blinded manner. In addition, formalin-fixed and paraffin embedded sections were subjected to antigen retrieval and incubated with mouse anti-mouse α-smooth muscle actin antibody (Sigma Aldrich MO851, Germany, A2547, 1:5000) as shown before [28]. Biotinylated donkey anti goat IgG and horse anti-mouse IgG (Santa Cruz, Germany) were used as secondary antibodies. Sections were incubated with ABC reagent, detected with DAB (Vectastain, Vector Laboratories, CA) and counterstained with hematoxylin. Digital images of the sections (n = 8 in each group) were superimposed on a grid, and the number of grid points overlapping collagen I fibers or α-smooth muscle actin fibers was recorded for each field. Twenty non-overlapping high-power fields at x400 magnification were analyzed in a blinded fashion. Data were expressed as the mean score ± SEM per 20 high power fields.

## Western immunoblotting

Kidneys of UUO and control mice were harvested on 7 and 14 days of life (n = 3 in each group) as described previously [5]. Neonatal kidneys were homogenized in protein lysis buffer (Tris 50 mM, $Na_4P_2O_3$ 1 mM, 2% SDS) containing protease inhibitor cocktail (Roche, Switzerland, #1836153). The protein content of the supernatants was measured using the BCA Protein Assay Kit (Pierce Biotechnology, MA, #23225). 20 μg of protein were separated on polyacrylamide gels at 160 V for 45 min and blotted onto nitrocellulose membranes (0,1 A per gel, 120 min). After blocking antibody-specific for 2 h in Tris-buffered saline with Tween-20 containing 5% nonfat dry milk and/or BSA, blots were incubated with primary antibodies 2 h at room temperature or at 4˚C overnight. TLR2 antibody (ThermoFisher Scientific, MA, #MA5-32787, 1:1000), Caspase-8 antibody (Cell Signaling Technology, MA, #4927, 1:1000), Bcl-2 antibody (Santa Cruz, Germany, sc7382, 1:200), Bax antibody (Cell Signaling Technology, MA, #27725, 1:1000), GSDMD antibody (Cell Signaling Technology, MA, #39754, 1:1000), GSDME antibody (Abcam, UK, ab215191, 1:500), HMGB1 antibody (Abcam, UK, ab18256, 1:1000), TNF antibody (Cell Signaling Technology, MA, #3707, 1:500), Galectin-3 antibody (Santa Cruz, Germany, sc19283, 1:500), α-SMA antibody (Sigma Aldrich, Germany, A2547, 1:5000), MMP-2 antibody (Santa Cruz, Germany, sc10736, 1:1000), TGF-β (Cell Signaling Technology, MA, #3711, 1:1000) were used for western blot analysis. GAPDH (DUNN Labortechnik, Germany, H86540M, 1:40000) was used as an internal loading control and to normalize samples. Blots were washed with Tris-buffered saline with Tween-20 and incubated with horseradish peroxidase-conjugated secondary antibody for 1 h at room temperature. Immune complexes were detected using enhanced chemiluminescence method. Blots were exposed to x-ray films (Kodak, Germany), the films were scanned, and protein bands were quantified using the densitometry program Image J. Each band represents one single neonatal mouse kidney. The uncropped gel images can be found in S1 Raw images.

## Statistical analysis

Data are presented as mean ± standard error. Comparisons between groups were made using one-way analysis of variance followed by the Student-Newman-Keuls test. Comparisons between left and right kidneys were performed using the Students t-test for paired data. Statistical significance was defined as $p < 0.05$.

## Results

### Neonatal UUO induces protein expression of TLR2

To measure the protein expression of TLR2 after UUO, we performed a western blot analysis of UUO and sham-operated kidneys of neonatal WT mice. Following UUO renal TLR2 protein expression increased significantly on day 14 of life in comparison to sham-operated controls (Fig 1A). We observed that as a response to the injury. TLR2 expression levels in neonatal

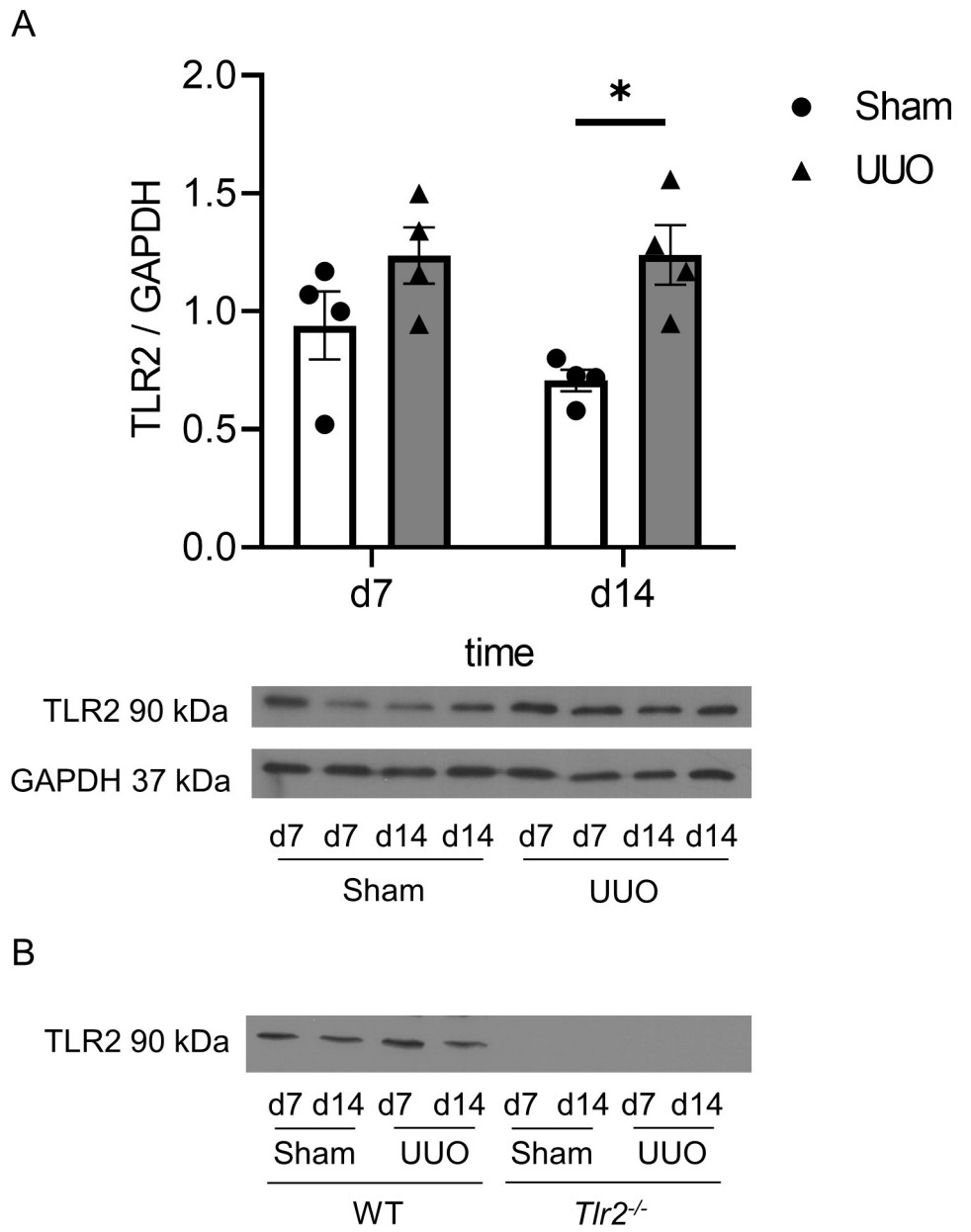

**Fig 1. Neonatal unilateral ureteral obstruction induces the expression of TLR2.** Neonatal WT mice were subjected to UUO or sham operation and their kidneys were harvested on d7 and d14. Lysates of whole kidneys were applied to SDS PAGE and consequent western blot analyses. The TLR2 expression level was significantly higher in UUO kidneys as compared to sham-operated controls on d14 (A). $Tlr2^{-/-}$ mice did not express TLR2 (B). n = 4; *p<0,05. Data are presented as mean +/- SEM.

WT-kidneys were increased. This analysis was also used to confirm that *Tlr2*<sup>-/-</sup> mice indeed did not express TLR2 (Fig 1B).

## TLR2 mediates tubular and interstitial apoptosis in neonatal kidneys with UUO

We next investigated tubular and interstitial apoptosis in neonatal kidneys from *Tlr2*<sup>-/-</sup> and WT mice having undergone UUO using TUNEL staining. Tubular apoptosis increased significantly in the obstructed kidneys at day 7 and 14 of life (Fig 2A–2C). TUNEL positive cells were mainly present in dilated distal tubules of the neonatal kidney. *Tlr2*<sup>-/-</sup> mice showed less tubular apoptosis compared to WT (Fig 2A–2C). Tubular apoptosis in *Tlr2*<sup>-/-</sup> mice was reduced on day 7 and day 14 of life by 41% and 30%, respectively (Fig 2C). Interstitial apoptosis increased following UUO and *Tlr2*<sup>-/-</sup> mice showed less interstitial apoptosis compared to WT (Fig 2D). Interstitial apoptosis in *Tlr2*<sup>-/-</sup> kidneys after UUO was reduced on day 7 and day 14 by 41%

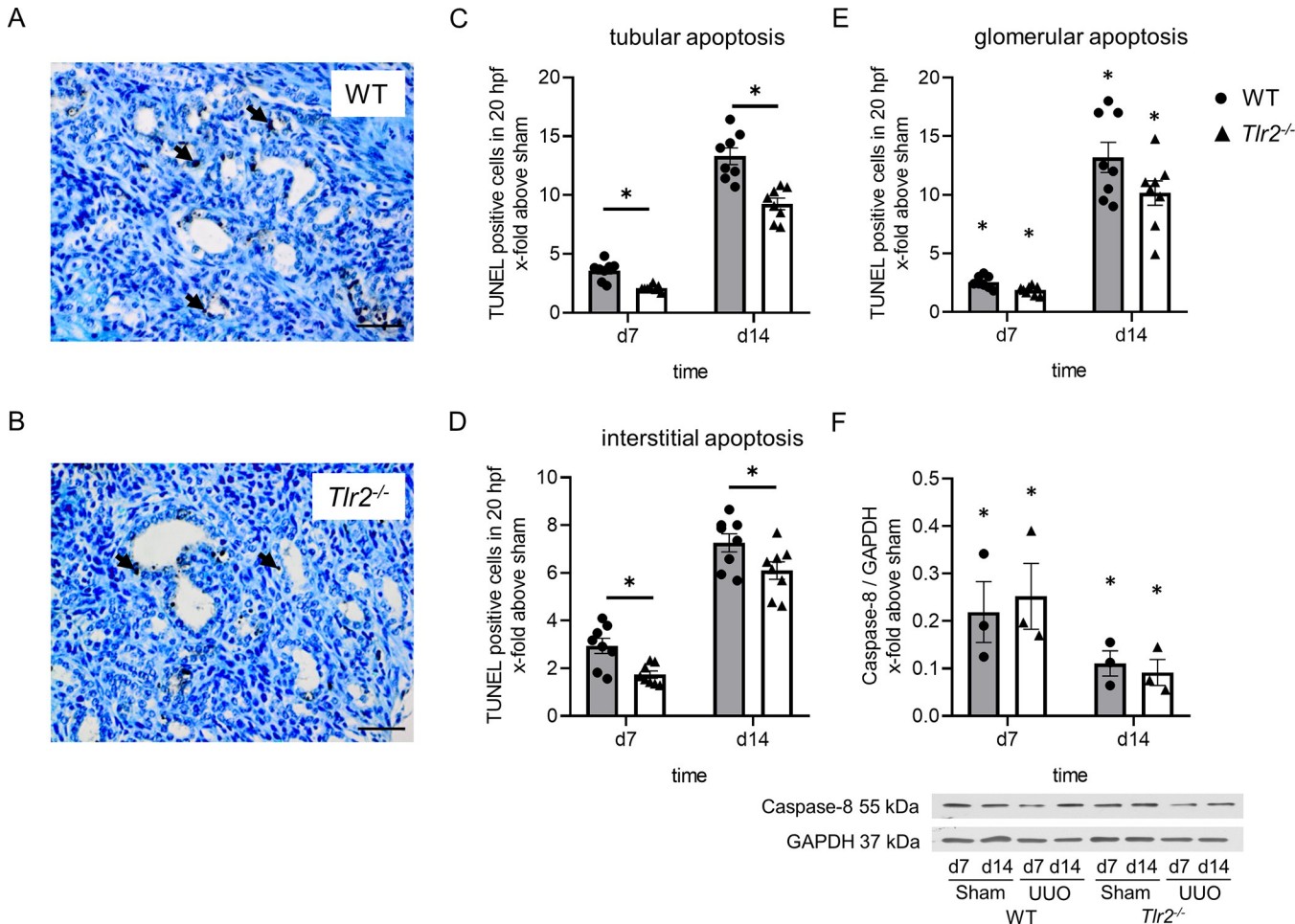

**Fig 2. Tubular and interstitial apoptosis in *Tlr2*<sup>-/-</sup> mice with UUO.** UUO was performed on the second day of life. Apoptotic cells were detected by TUNEL staining in sections. TUNEL-positive cells in WT (A) and *Tlr2*<sup>-/-</sup> (B) neonatal mouse kidneys appeared predominantly in distal tubules and in the interstitium. Arrows indicate tubular apoptotic cells. Quantification indicates significant decreases of numbers of TUNEL-positive tubular (C) and interstitial (D) cells in *Tlr2*<sup>-/-</sup> as compared to WT UUO kidneys. The number of apoptotic nuclei in glomeruli did not differ between *Tlr2*<sup>-/-</sup> and WT specimen (E); n = 8. Whole kidneys were lysed for western blot analyses (F). Caspase-8 expression was reduced in UUO-kidneys indicating apoptosis following ureteral obstruction. Significant differences between WT and *Tlr2*<sup>-/-</sup> were not observed (F). Results are indicated as x-fold relative to sham-operated controls. n = 3 (Western Blot); *p<0,05. Data are presented as mean +/- SEM. Bar = 100µm. Standalone * represents significant differences between Sham and UUO results.

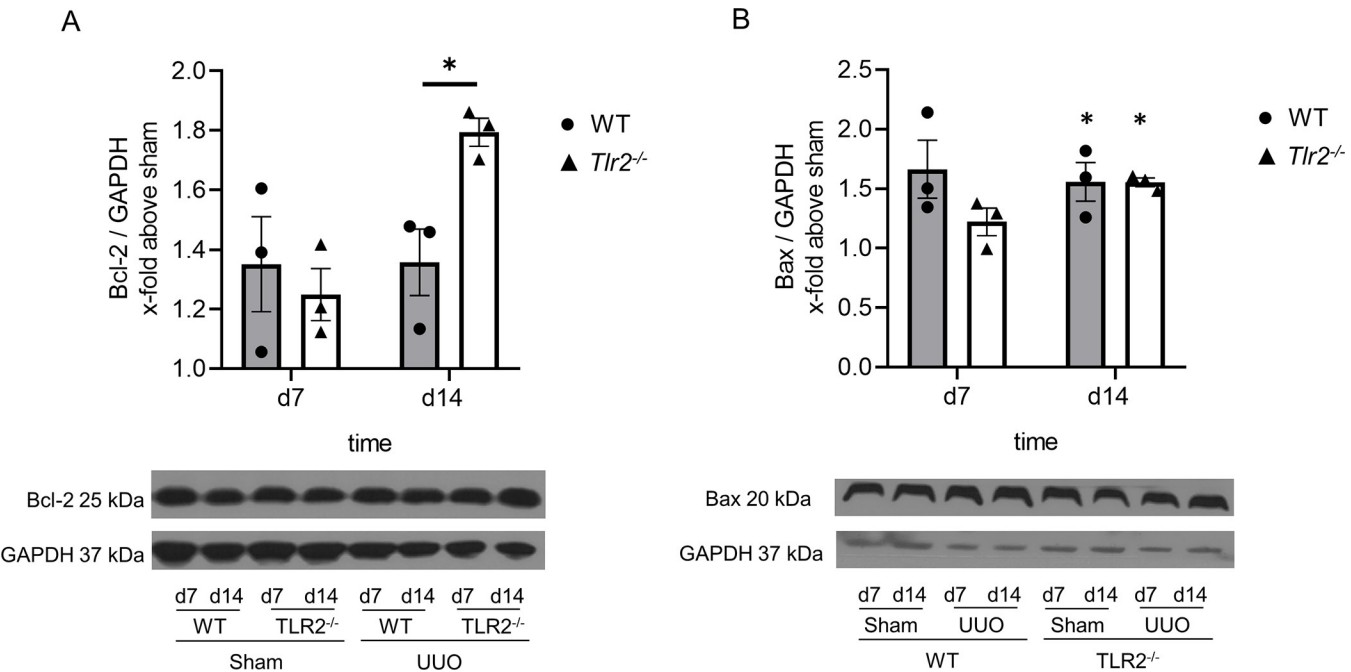

**Fig 3.** *Tlr2*$^{-/-}$ **mice show increased renal Bcl-2 expression in comparison to WT following UUO.** Neonatal mice were subjected to UUO or sham operation. Western blot analysis was performed at day 7 and 14 of life. UUO induced the expression of Bcl-2 in *Tlr2*$^{-/-}$, but not in WT kidneys at day 14 (A). UUO induced expression of Bax in the neonatal kidney without significant differences between WT and *Tlr2*$^{-/-}$ (B). Results are indicated as x-fold relative to sham-operated controls. n = 3; *p<0,05. Data are presented as mean +/- SEM. Standalone * represents significant differences between Sham and UUO results.

and 16%, respectively (Fig 2D). Glomerular apoptosis increased following UUO, but without significant differences between *Tlr2*$^{-/-}$ and WT kidneys (Fig 2E). Apoptosis was also measured by caspase-8 protein expression using western blot (Fig 2F). Cleavage of caspase-8 indicates apoptosis [29]. Both WT and *Tlr2*$^{-/-}$ mice showed reduced caspase-8 expression indicating apoptotic cell death after UUO (Fig 2F). However, caspase-8 expression was not significantly different between WT and *Tlr2*$^{-/-}$ mice (Fig 2F), which may be explained by the lack of compartment-specific analysis of the neonatal kidney. For further analysis of cell death in our model we analyzed the anti-apoptotic marker Bcl-2 and the pro-apoptotic marker Bax using western blot (Fig 3). Neonatal *Tlr2*$^{-/-}$ mice showed a higher expression of Bcl-2 at day 14 in comparison to WT (Fig 3A), confirming that TLR2 mediates apoptosis in the neonatal model of obstructive nephropathy. Bax expression increased following UUO at day 14, without significant differences between *Tlr2*$^{-/-}$ and WT kidneys.

## Pyroptosis generally increased after neonatal UUO, yet its grades in WT are indistinguishable from those of *Tlr2*$^{-/-}$ murine specimen

To measure the potential impact of TLR2 expression on pyroptosis upon UUO versus samples from sham-operated mice, abundance of cleaved GSDMD and full-length GSDME expression in respective kidneys at day 7 and 14 of life were measured by immunoblotting (Fig 4A and 4B). While the abundance of cleaved GSDMD increased due to the obstruction in samples of both phenotypes at d14 (Fig 4A), that of full-length GSDME decreased after UUO (Fig 4B) as if pyroptosis became operative upon UUO. However, WT and *Tlr2*$^{-/-}$ mice borne specimen were undistinguishable in this regard (Fig 4A and 4B). Thus, TLR2 is not involved in pyroptosis after ureteral obstruction in the neonatal kidney. Additionally, we analyzed the expression of the pyroptosis markers HMGB1 and TNF in UUO- and sham-operated neonatal kidneys at

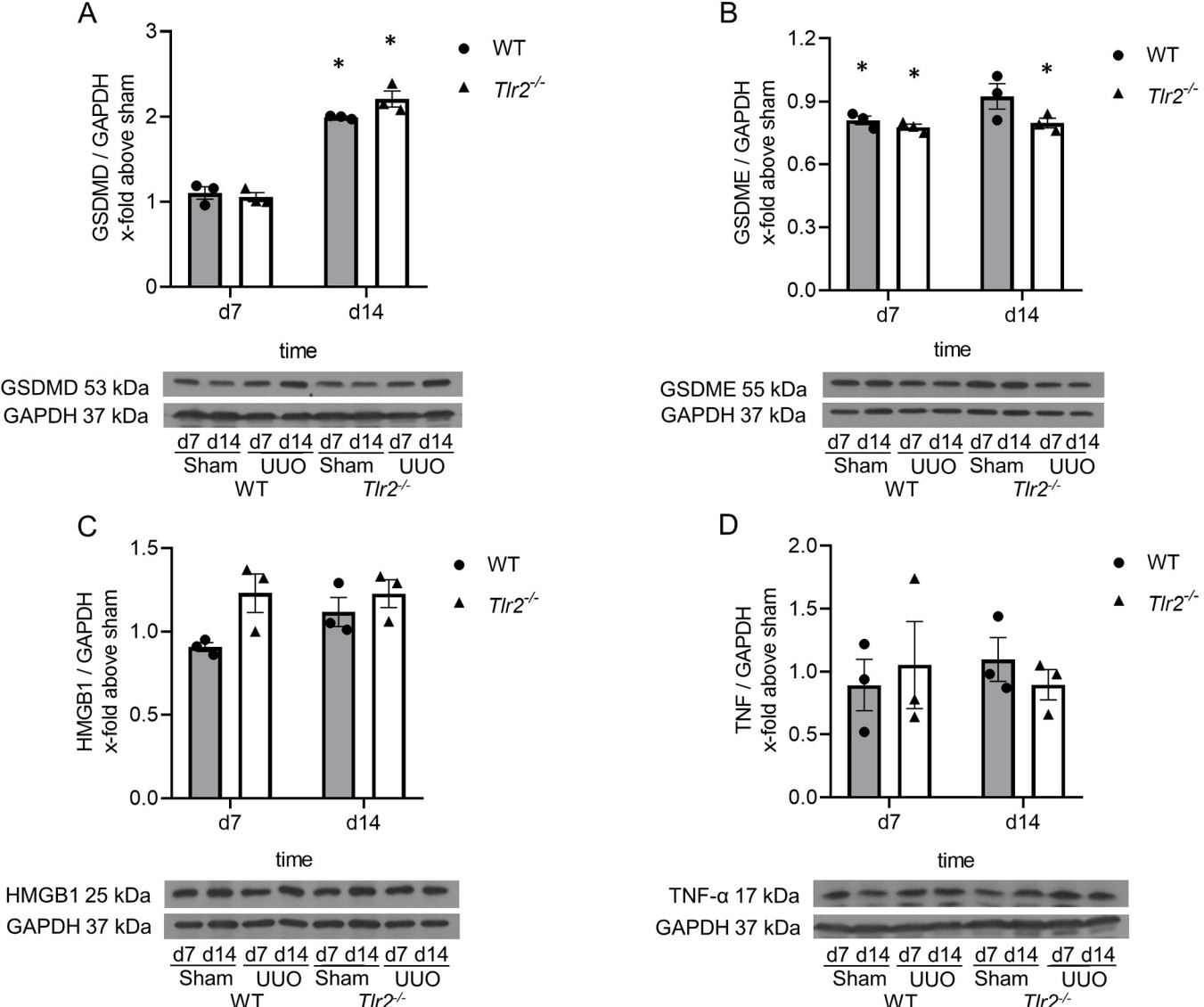

**Fig 4. Pyroptosis increases after UUO.** Neonatal mice were subjected to UUO or sham operation. Whole kidneys were processed for western blot analysis at day 7 and 14 of life. UUO induced cleaved GSDMD expression (A) and full-length GSDME cleavage (B), but with no significant differences between WT and *Tlr2*-/-. Pyroptosis markers HMGB1 (C) and TNF-α (D) did not show significant differences between WT and *Tlr2*-/-. Results are indicated as x-fold relative to sham-operated controls. n = 3; *p<0,05. Data are presented as mean +/- SEM. Standalone * represents significant differences between Sham and UUO results.

day 7 and 14 of life (Fig 4C and 4D). Expression levels of both proteins remained constant throughout the analysis period and were indistinguishable in specimen of both genotypes (Fig 4C and 4D). We conclude that TLR2 has no impact on pyroptosis, HMGB1 and TNF expression in the neonatal kidney having been subjected to UUO.

## Proliferation decreased and tubular atrophy increased in neonatal kidneys after UUO, without significant differences between *Tlr2*-/- and WT mice

Proliferation in neonatal *Tlr2*-/- and WT mouse kidneys was measured using Ki67 staining in obstructed and sham-operated kidneys at day 7 and 14 of life (Fig 5A–5C). Following obstruction, proliferation decreased significantly, but without a significant difference between WT

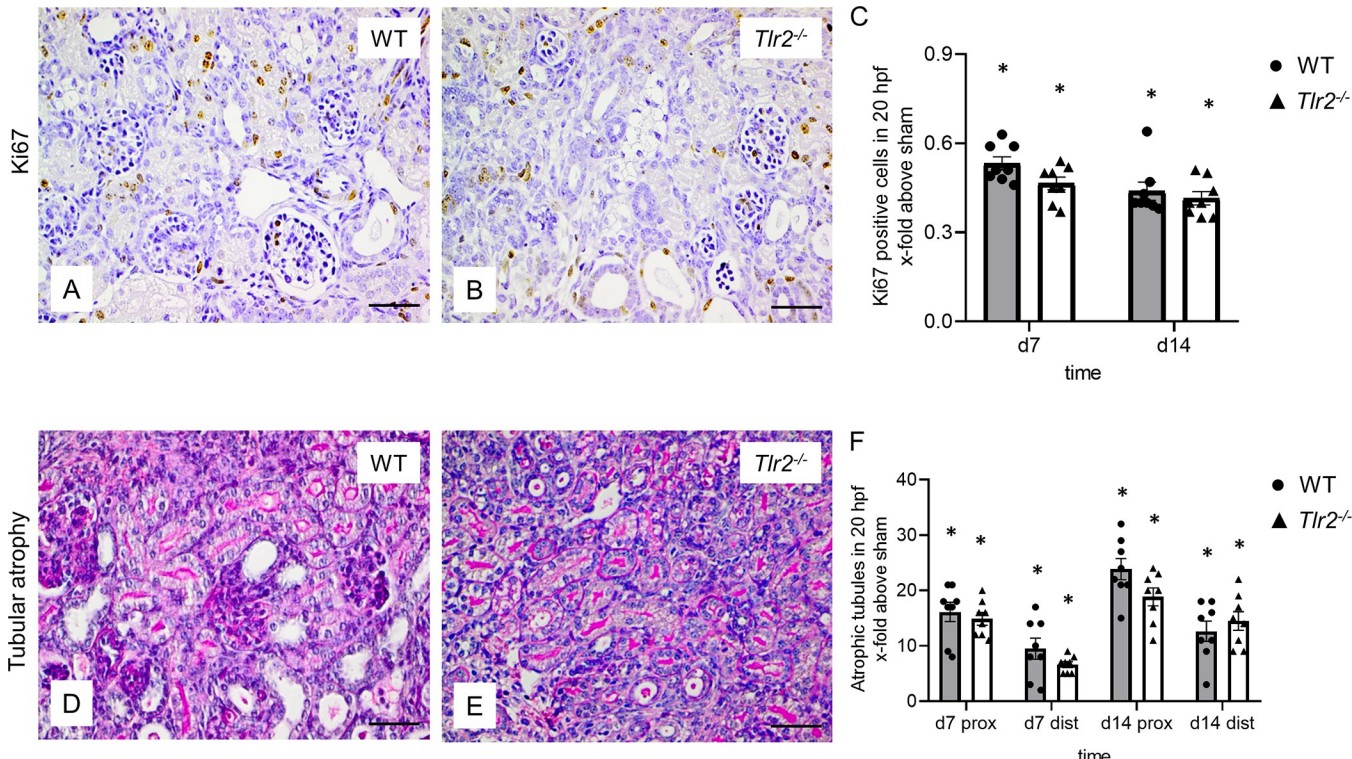

**Fig 5. Analysis of tubular proliferation and atrophy in neonatal mice following UUO.** Immunohistochemical staining of Ki67 (A and B) and proximal (prox) and distal (dist) atrophic tubules (D and E) in neonatal UUO WT (A and D) and *Tlr2*⁻/⁻ (B and E) mice at day 7. Quantification of Ki67-positive tubular cells in UUO-kidneys (C) shows a decrease following UUO without significant differences between WT and *Tlr2*⁻/⁻. Quantitative analysis of tubular atrophy in UUO-kidneys on day 7 and day 14 (F) shows an increase of tubular atrophy in proximal and distal tubules following UUO, but no significant differences between WT and *Tlr2*⁻/⁻ mice in neither compartment. Results are indicated as x-fold increase above sham operated control in 20 hpfs; n = 8. Data are presented as mean +/- SEM. Bar = 100µm. Standalone * represents significant differences between Sham and UUO results.

and *Tlr2*⁻/⁻ mice (Fig 5C). To measure tubular atrophy, a Periodic Acid Schiff (PAS) staining of neonatal WT and *Tlr2*⁻/⁻ kidneys was performed after UUO at day 7 and 14 of life (Fig 5D–5F). Tubular atrophy increased at d7 and d14 in UUO kidneys and was more prominent in proximal than distal tubules following UUO. Between WT and *Tlr2*⁻/⁻ mice no significant differences in tubular atrophy could be observed (Fig 5F). We conclude that TLR2 does not influence proliferation or tubular atrophy in the neonatal kidney with UUO.

## M2 macrophage infiltration, T-lymphocyte infiltration and fibrosis increased after UUO but were not different between *Tlr2*⁻/⁻ and WT mice

Influence of TLR2 on M2 macrophage infiltration in neonatal *Tlr2*⁻/⁻ and WT mouse kidneys with UUO was measured using galectin-3 staining (Fig 6A–6C) and galectin-3 protein expression using western blot (Fig 7A). Increased expression of interstitial galectin-3 is a feature of the regenerative anti-inflammatory alternative (M2) macrophage phenotype. UUO induced a vast M2 macrophage infiltration in the interstitium of neonatal *Tlr2*⁻/⁻ and WT kidneys, 10-fold at d14, but without significant differences between the two lines (Fig 6A–6C). UUO induced a marked galectin-3 expression in the neonatal kidneys but was not different between *Tlr2*⁻/⁻ and WT mice (Fig 7A). T-lymphocyte infiltration was assessed by CD3 staining (Fig 6D–6F). UUO induced CD3 positive T-cell-infiltration in neonatal kidneys of WT and *Tlr2*⁻/⁻ mice, at d7 and d14 (Fig 6D–6F). No significant differences between WT and *Tlr2*⁻/⁻ kidneys

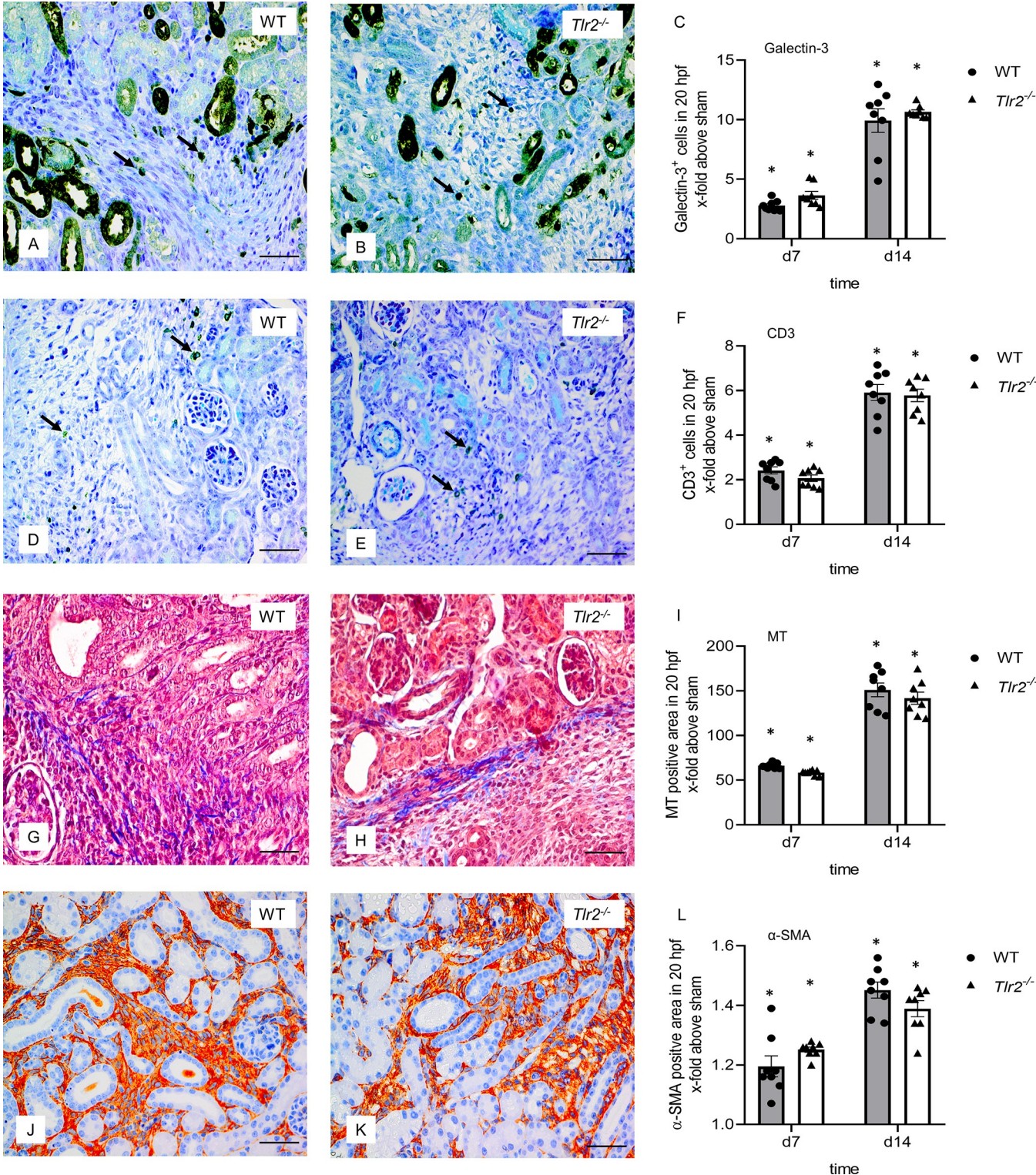

**Fig 6. Immune cell infiltration and fibrosis in neonatal UUO kidneys.** Immunohistological staining for galectin-3, a M2 macrophage marker, of WT (A) and *Tlr2*⁻/⁻ (B) mice on day 14. Quantitative analysis shows that UUO induced interstitial galectin-3 positive macrophage infiltration (arrow) in neonatal kidneys (C), but without significant differences between WT and *Tlr2*⁻/⁻ mice. Immunohistological staining for CD3 (arrow), a T-lymphocyte marker, of WT (D) and *Tlr2*⁻/⁻ (E) mice on day 14. Quantitative analysis shows CD3 positive T-cell infiltration in neonatal kidneys following UUO (F) without significant differences between WT and *Tlr2*⁻/⁻ mice. Renal sections of UUO- and sham-operated were stained for Masson's Trichrome (MT) at 7 and 14 days of life. UUO induced collagen deposition in neonatal kidneys of WT (G) and *Tlr2*⁻/⁻ mice (H) on day 14. Renal sections of UUO- and sham-operated mice were stained for α-SMA at

7 and 14 days of life. UUO induced α-SMA expression in neonatal kidneys of WT (J) and *Tlr2-/-* mice (K) on day 14. Analysis of α-SMA positive myofibroblasts in UUO-kidneys on day 7 and 14 (L) showed no significant differences between the two groups. Results are indicated as x-fold increase above sham operated control in 20 hpfs (x400); n = 8. Data are presented as mean +/- SEM. Bar = 100μm. Standalone * represents significant differences between Sham and UUO results.

could be observed (Fig 6F). We conclude that TLR2 does not have an impact on the M2 macrophage and T-lymphocyte infiltration in the neonatal kidney with UUO. To study interstitial fibrosis in WT and *Tlr2-/-* mice after neonatal UUO, Masson's Trichrome and α-smooth muscle actin (α-SMA) staining of kidney sections were performed (Fig 6G–6L). Additionally, protein expression of α-SMA, matrix metalloproteinase-2 (MMP-2), and transforming growth factor (TGF)-β was measured by western blot (Fig 7B–7D). Interstitial collagen deposition

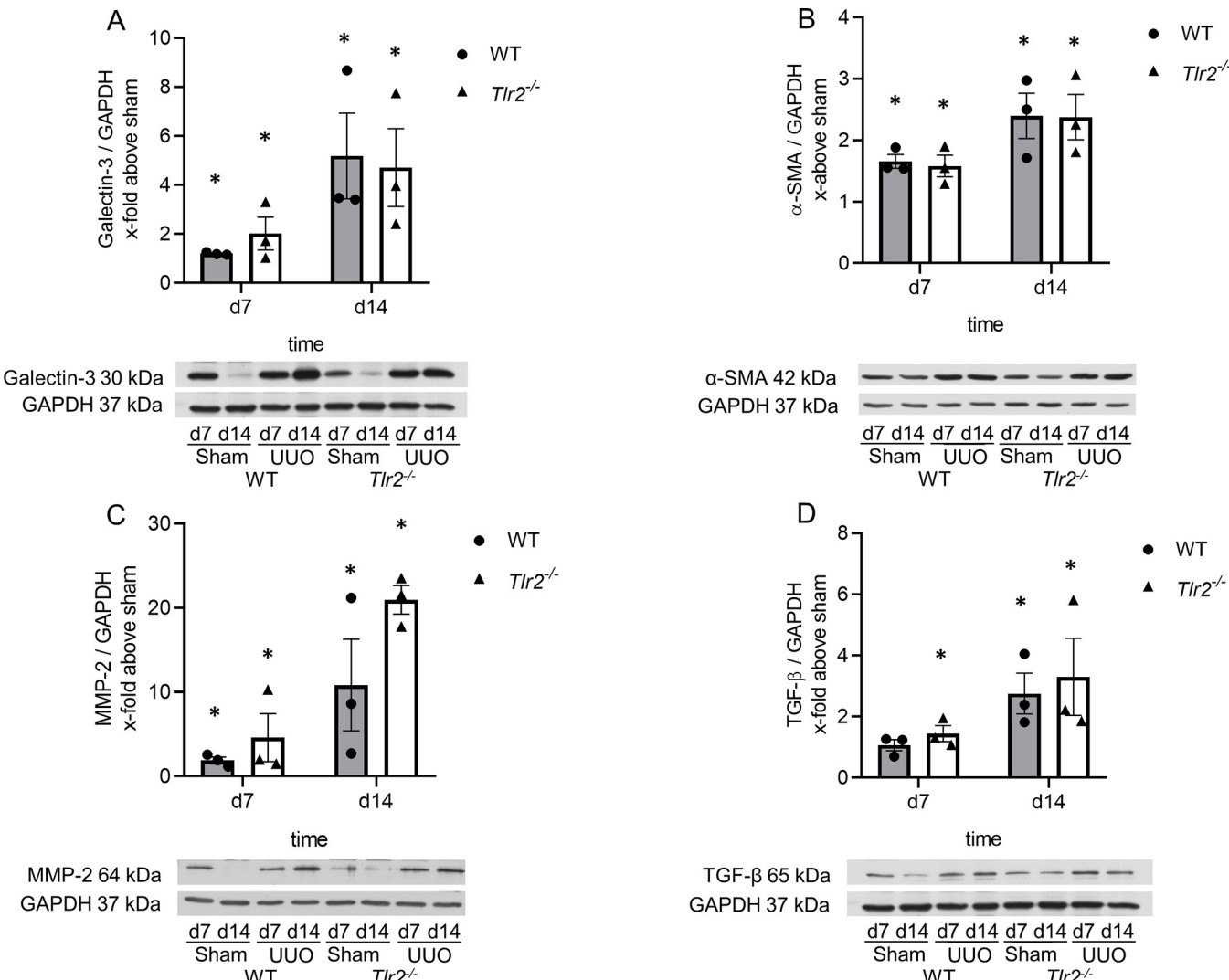

**Fig 7. Protein expression of macrophage and fibrosis markers after UUO.** Neonatal mice were subjected to UUO or sham operation. Whole kidneys were processed for western blot analysis at day 7 and 14. UUO induced galectin-3 expression in neonatal UUO kidneys, without significant differences between WT and *Tlr2-/-* kidneys (A). UUO induced the expression of the fibrotic markers α-SMA (B), MMP-2 (C) and TGF-β (D), but without significant differences between *Tlr2-/-* and WT mice at day 7 and 14 of life. Expression is indicated as x-fold increase above sham operated control, n = 3. Data are presented as mean +/- SEM. Standalone * represents significant differences between Sham and UUO results.

measured by the Masson's Trichrome staining, increased after UUO in both WT and *Tlr2*<sup>-/-</sup> mice (Fig 6G–6I). The abundance of α-SMA increased at d7 and d14 after UUO but was not different in *Tlr2*<sup>-/-</sup> mice in comparison to WT mice (Fig 6J–6L). Following neonatal UUO, α-SMA expression increased significantly at d7 and d14 (Fig 7B), but without significant differences between WT and *Tlr2*<sup>-/-</sup> kidneys. MMP-2 expression increased significantly after ureteral obstruction at d7 and d14 in WT and *Tlr2*<sup>-/-</sup> mice (Fig 7D), with a slightly increasing trend for *Tlr2*<sup>-/-</sup> kidneys, but without significant differences between the two lines. UUO induced increased TGF-β-expression (Fig 7D) in the neonatal kidneys, but *Tlr2*<sup>-/-</sup> mice were not significantly different from WT mice. We therefore conclude that TLR2 does not attenuate renal fibrosis in the neonatal kidney with UUO.

## Discussion

Our study indicates involvement of TLR2 in the mediation of apoptosis in the early life developing murine kidney suffering from ureteral obstruction. This result is of potential relevance because the pattern recognition receptor TLR2 is an element of innate immunity, which drives various kidney diseases elicited experimentally in animal models [8, 30]. Exemplarily, TRL2 induces inflammation and renal injury in the adult model of ischemia/reperfusion injury and in streptozotocin-induced diabetic mice [13, 17, 31].

Here, we investigated a potential pro-apoptotic role of TLR2 using neonatal *Tlr2*<sup>-/-</sup> mice we subjected to UUO. Firstly, we show that TLR2 expression is upregulated after UUO in neonatal mouse kidneys, which indicates a potential role of this receptor in obstructive nephropathy. This is in line with studies in adult mice, where TLR2 expression is markedly upregulated after UUO [32].

Secondly, by using neonatal *Tlr2*<sup>-/-</sup> mice we demonstrate that TLR2 mediates tubular and interstitial apoptosis in the obstructed neonatal kidney. Following UUO, neonatal kidneys of *Tlr2*<sup>-/-</sup> mice displayed markedly reduced abundance of tubular apoptotic nuclei, measured by a TUNEL assay. These apoptotic nuclei were predominantly present in distal tubular cells, which is in line with our published data on programmed cell death in the neonatal kidney [5]. In addition, interstitial apoptosis was also markedly reduced in neonatal *Tlr2*<sup>-/-</sup> mouse kidneys, suggesting their carriage of either less apoptotic infiltrating leukocytes or less apoptotic myofibroblasts and fibroblasts as compared to controls. In contrast, glomerular apoptosis did not differ between *Tlr2*<sup>-/-</sup> and WT kidneys. To potentially reinforce differences in tubular and interstitial apoptosis in WT and *Tlr2*<sup>-/-</sup> mice borne kidneys, caspase 8 expression was measured, but showed no difference between the lines in the neonatal kidneys of newborn mice undergoing UUO. Caspase-8 expression has limitations as an apoptotic marker, as it is a marker of the early phase of apoptosis [33]. In contrast, TUNEL staining has been designed to detect apoptotic cells that undergo extensive DNA degradation during the late stages of apoptosis [34]. Therefore, the differences between *Tlr2*<sup>-/-</sup> and WT kidneys may not be detectable during the initiation phase of apoptosis but become apparent in the late apoptotic stages of DNA degradation. In addition, as caspase-8 protein expression is measured for all compartments of the kidney by using whole neonatal kidney lysates for western blot analysis, a compartment-restricted expression of caspase 8 might have indicated differences we missed by our global approach. As our results were conflicting, we decided to include Bcl-2 as an anti-apoptotic marker [35, 36]. The increase of Bcl-2 expression in *Tlr2*<sup>-/-</sup> kidneys further confirms our findings that TLR2 mediates apoptosis in neonatal UUO. Our result "apoptosis reduction in *Tlr2*<sup>-/-</sup> mice" is in line with Leemans et al., who showed that TLR2 activates the apoptotic pathway in UUO-kidneys of adult mice [32]. The authors measured apoptosis through staining of active caspase-3 in cells, which was significantly reduced in *Tlr2*<sup>-/-</sup> mice after 7 days of ureteral

obstruction. Unfortunately, we were not able to measure caspase-3 in neonatal kidneys most certainly as the abundance of this antigen was below the detection limit. In the neonatal model of unilateral ureteral obstruction, it is currently unknown how this *Tlr2*[-/-] associated apoptotic pathway is activated, caspase-3 might not be involved at all. TLR2, as a part of the innate immune system is activated by bacterial lipoproteins and signals for apoptosis through MyD88 via a pathway involving Fas-associated death domain protein and caspase-8 [37, 38]. Inflammation following UUO is not mediated by bacteria. UUO induces sterile inflammation, which is mediated by DAMPs, of which a variety might be able to activate the TLR2 mediated apoptotic pathway [23, 39]. High levels of apoptosis are associated with nephron loss in the developing kidney and thus the loss of renal function [40].

Besides apoptosis, necrosis and regulated necrosis are cell death mechanisms that are operative during UUO [41]. Previously we demonstrated necroptosis, a form of regulated necrosis, to be increased in the neonatal kidney undergoing UUO [5]. Here we demonstrate for the first time, that pyroptosis is also upregulated in the neonatal kidney with UUO. Pyroptosis is a gasdermin-mediated programmed cell death that involves the activation of inflammasomes by DAMPs [42]. Pyroptosis plays an important role in the progression of kidney disease and is involved in various kidney disease models [43]. Here we analyzed pyroptosis in neonatal *Tlr2*[-/-] and WT mice with UUO by the analyzing cleaved GSDMD and full-length GSDME abundances. The cleavage of GSDMD, or alternatively GSDME is a crucial step in the initiation of pyroptosis and pore formation [24]. In our study we show that UUO in neonatal kidneys induces. This observation is in accordance with findings in adult mice and rats subjected to UUO [44, 45]. However, pyroptosis was not different in *Tlr2*[-/-] mice as compared to WT controls suggesting that TLR2 is not involved in pyroptotic cell death following neonatal UUO.

TLR2 borne intracellular signal transduction is induced by sterile insult and triggers inflammation [46]. Inflammation is a major driver of UUO pathology associated with release of DAMPs and pro-inflammatory cytokines initiating it [23, 39]. It is currently unknown what DAMP induces apoptosis through TLR2. Numerous DAMPs are putative ligands of TLR2 [23, 47]. HMGB1 is involved in inflammasome activation as well as regulation of apoptosis [48, 49]. To investigate if HMGB1 mediates sterile inflammation in neonatal kidneys after UUO, we measured kidney borne HMGB1 expression levels. We show here for the first time that HMGB1 expression in neonatal kidneys did not increase after UUO. This contrasts with findings in kidneys of adult mice, where UUO caused a marked upregulation of kidney inherent HMGB1 [50] and may be explainable by differential expression in the course of kidney development. Nephrogenesis in mice starts at embryonic day 8 and is completed 2–3 weeks after birth. In contrast to adult UUO, neonatal UUO impairs kidney development and reduces nephron mass, as nephrogenesis is still going on. Thus, HMGB1 signaling may be differentially regulated in neonatal and adult mice with UUO. TNF, a pro-inflammatory cytokine, is another mediator of sterile inflammation following UUO. The activation of TLR2 principally leads to production of TNF [13, 51]. TNF is highly upregulated in kidneys of adult mice with UUO [52]. Contrary to our expectations, *Tlr2*[-/-] and WT mice did not upregulate TNF expression after UUO. Additionally, there were no differences between kidneys of the genotypes. Thus, sterile inflammation in neonatal kidneys after UUO seems to be mediated by neither HMGB1 nor TNF. Identification of DAMPs and mediators eliciting apoptosis in neonatal kidneys through TLR2 still is to be investigated.

In order to examine if TLR2 has an influence on proliferation in the neonatal UUO model, we analyzed relative abundances of KI67 positive tubular cells. We show that the number of proliferating tubules decreased following UUO. Whereas in adult UUO a decrease of proliferation was observed in *Tlr2*[-/-] kidneys in comparison to WT [32], in neonatal UUO lacked such

dichotomy. Since neonatal kidney cells highly proliferate in general, slight differences between the two lines could have been overshadowed by the impact UUO has on nephrogenesis.

Morphological alterations in the tubular compartment play an important role in the pathogenesis of neonatal obstructive uropathy. Tubular atrophy is mainly a concern in proximal tubules, which is in line with previous results [5]. We were not able to observe differences between *Tlr2*⁻/⁻ and WT neonatal kidneys regarding tubular atrophy. TLR2 does not influence tubular atrophy after neonatal UUO.

Renal interstitial fibrosis develops parallel to renal injury and sterile inflammation after neonatal UUO. Fibroblast density increases due to local proliferation of resident fibroblasts, the recruitment of fibrocytes and possibly epithelial-mesenchymal transition (EMT) [53]. T-lymphocytes and M2 macrophages are crucial in the development of renal fibrosis [54, 55]. It has been shown that in the adult UUO model M2 macrophages facilitate renal fibrosis [56]. Several studies have shown that depletion of T-cells after UUO in adult mice results in a reduction of renal fibrosis [57, 58]. Here, we observed a marked infiltration of M2-macrophages and T-cells after neonatal UUO. However, for all measured parameters there were no significant differences between *Tlr2*⁻/⁻ and WT neonatal mice. This is in line with published data showing that a TLR2 knockout does not influence macrophage infiltration in adult UUO [32]. Expansion of fibrous connective tissue and abnormal deposition of extracellular matrix produced by myofibroblasts build the basis for fibrotic diseases. We measured the quantity of fibrotic collagen fibers in neonatal UUO kidneys, evaluated myofibroblasts by α-SMA staining and protein expression, and investigated the fibrotic marker TGF-β. The results show a significant increase of renal fibrosis in neonatal kidneys after UUO. We were not able to observe differences between *Tlr2*⁻/⁻ and WT mice. Matrix metalloproteinases are involved in EMT following UUO. MMP-2 aggravates the expression of EMT-associated molecules and renal fibrosis in adult UUO [59]. In our study, MMP-2 increased noticeably after neonatal UUO, but without differences between *Tlr2*⁻/⁻ and WT mice. By contrast, adult UUO kidneys showed a diminished expression of MMP-2 in *Tlr2*⁻/⁻ mice. Thus, expression of MMP-2 after UUO may be differently regulated in neonatal and adult kidneys. Overall, TLR2 does not influence interstitial fibrosis in neonatal mouse kidneys with UUO.

Numerous studies showed involvement of TLR2 in acute kidney injury [13, 60]. Inhibition of TLR2 reduced the recruitment of NK cells, as well as neutrophil infiltration and renal damage to the kidney after IRI [60, 61]. TLR2 and its endogenous stress ligands are markedly upregulated in obstructed kidneys in adult mice [32, 62]. Our results demonstrate that TLR2 does play an essential role neither in kidney inflammation nor in the development of renal fibrosis following neonatal UUO. Recently, it has been shown that inhibition of both RAS and TLR2 has an additive ameliorative effect on UUO injury of the kidney [63]. Given this information it may be more effective to target additional pathways besides TLR2 in the obstructed kidney to ameliorate inflammation and fibrosis.

## Conclusion

TLR2 plays an important role in mediating tubular and interstitial apoptosis in the neonatal kidney with obstruction. Inhibition of TLR2 in obstructive nephropathy could prevent apoptosis and save nephron mass, which would be otherwise irreversibly lost. Blocking TLR2 may be beneficial in the developing kidney with obstruction until the obstruction resolves or a surgical correction is performed. However, TLR2 does not influence inflammatory responses or development of renal fibrosis after UUO. Thus, a combination with other inhibitors may be of advantage.

## Supporting information

**S1 Raw images. Western blot raw images.** Uncropped western blot gel images for TLR2 and GAPDH in neonatal WT kidneys (on day 7 and 14 of life). * marks the section used in Fig 1. TLR2 and GAPDH were visualized separately, but they represent the same gel. Uncropped western blot gel images for TLR2 and GAPDH in neonatal WT kidneys (on day 7 and 14 of life). * marks the section used in Fig 1. For Bax Gel 2 was used. Uncropped western blot gel images for Caspase-8 and GAPDH in neonatal WT and TLR2-/- kidneys (on day 7 and 14 of life). * marks the section used in Fig 2. Caspase-8 and GAPDH were visualized separately, but they represent the same gel. Uncropped western blot gel images for GSDMD and GAPDH in neonatal WT and TLR2-/- kidneys (on day 7 and 14 of life). * marks the section used in Fig 3. GSDMD and GAPDH were visualized separately, but they represent the same gel. Uncropped western blot gel images for GSDME and GAPDH in neonatal WT and TLR2-/- kidneys (on day 7 and 14 of life). * marks the section used in Fig 3. GSDME and GAPDH were visualized separately, but they represent the same gel. Uncropped western blot gel images for HMGB1 and GAPDH in neonatal WT and TLR2-/- kidneys (on day 7 and 14 of life). * marks the section used in Fig 3. Uncropped western blot gel images for TNF-α and GAPDH in neonatal WT and TLR2-/- kidneys (on day 7 and 14 of life). * marks the section used in Fig 3. TNF-α and GAPDH were visualized separately (different exposer times), but they represent the same gel. Uncropped western blot gel images for Galectin-3, TGF-β, and GAPDH in neonatal WT and TLR2-/- kidneys (on day 7 and 14 of life). * marks the section used in Fig 6. Uncropped western blot gel images for α-SMA and GAPDH in neonatal WT and TLR2-/- kidneys (on day 7 and 14 of life). * marks the section used in Fig 6. Uncropped western blot gel images for MMP-2 and GAPDH in neonatal WT and TLR2-/- kidneys (on day 7 and 14 of life). * marks the section used in Fig 6. MMP-2 and GAPDH were visualized separately, but they represent the same gel.
(PDF)

## Author Contributions

**Conceptualization:** Carsten Kirschning.

**Data curation:** Maja Wyczanska.

**Formal analysis:** Maja Wyczanska, Jana Rohling.

**Investigation:** Ursula Keller.

**Methodology:** Maja Wyczanska, Jana Rohling, Marcus R. Benz.

**Project administration:** Bärbel Lange-Sperandio.

**Resources:** Bärbel Lange-Sperandio.

**Supervision:** Bärbel Lange-Sperandio.

**Validation:** Bärbel Lange-Sperandio.

**Writing – original draft:** Maja Wyczanska, Bärbel Lange-Sperandio.

**Writing – review & editing:** Maja Wyczanska, Bärbel Lange-Sperandio.

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
