## [Decision Letter · Decision Letter 0]

17 May 2023

PONE-D-23-12099TLR2 mediates renal apoptosis in neonatal mice subjected experimentally to obstructive nephropathyPLOS ONE

Dear Dr. Lange-Sperandio,

Thank you for submitting your manuscript to PLOS ONE. After careful consideration, we feel that it has merit but does not fully meet PLOS ONE’s publication criteria as it currently stands. Therefore, we invite you to submit a revised version of the manuscript that addresses the points raised during the review process.

We look forward to receiving your revised manuscript.

Kind regards,

Franziska Theilig

Academic Editor

PLOS ONE

Journal Requirements:

2. To comply with PLOS ONE submissions requirements, in your Methods section, please provide additional information regarding the experiments involving animals and ensure you have included details on (1) methods of sacrifice, and (2) efforts to alleviate suffering.

6. Please amend the manuscript submission data (via Edit Submission) to include author Dr. Ursula Keller.

7. PLOS ONE now requires that authors provide the original uncropped and unadjusted images underlying all blot or gel results reported in a submission’s figures or Supporting Information files. This policy and the journal’s other requirements for blot/gel reporting and figure preparation are described in detail at https://journals.plos.org/plosone/s/figures#loc-blot-and-gel-reporting-requirements and https://journals.plos.org/plosone/s/figures#loc-preparing-figures-from-image-files. When you submit your revised manuscript, please ensure that your figures adhere fully to these guidelines and provide the original underlying images for all blot or gel data reported in your submission. See the following link for instructions on providing the original image data: https://journals.plos.org/plosone/s/figures#loc-original-images-for-blots-and-gels. 

Reviewers' comments:

Reviewer's Responses to Questions

**Comments to the Author**

1. Is the manuscript technically sound, and do the data support the conclusions?

Reviewer #1: Partly

Reviewer #2: Yes

2. Has the statistical analysis been performed appropriately and rigorously? 

Reviewer #1: Yes

Reviewer #2: Yes

3. Have the authors made all data underlying the findings in their manuscript fully available?

Reviewer #1: Yes

Reviewer #2: Yes

4. Is the manuscript presented in an intelligible fashion and written in standard English?

Reviewer #1: Yes

Reviewer #2: Yes

5. Review Comments to the Author

Reviewer #1: In this manuscript, a series of studies concerning the important role of TLR2 in the neonatal kidney with obstruction have been showed. In this study, TLR2’s crucial role in mediating tubular and interstitial apoptosis is highlighted, which, as authors mentioned in the manuscript, may shade light on developing new treatment approaches to congenital obstructive nephropathy in the future. It is worth valuing for the efforts made by the researchers, however, there are several points need to be addressed.

1. The subject of this manuscript is TLR2 mediates renal apoptosis in neonatal mice subjected experimentally to obstructive nephropathy. However, in this manuscript, the proportion of narrative and experimental evidence of renal apoptosis is low, while other results that do not differ are slightly more discussed, indicating a shift in emphasis in the article. Authors could increase the types of experiments on renal apoptosis or dig deeper into the mechanism of TLR2 on renal apoptosis to make full-text logic more clarity.

2. Result I and II are both results about the center of the article, which is renal apoptosis, but they have somewhat smaller sample sizes, even less than the sample size of the undifferentiated results. It is recommended that researchers increase the sample size to make the article data more convincing.

3. The method section of the article does not describe the source of the knockout mice, and it is suggested to add a description to increase the credibility of the data. Moreover, the references of the article are old, it is recommended to use the journal literature within the last 3 years or 5 years if possible.

Reviewer #2: 1) In Figure 1, the expression of TLR2 on d14 was less than d7 in the sham group, and in the UUO group the TLR2 expression on d7 and d14 was similar. How to explain this result?

2) In figure 2F and 3, there were 8 groups, but histogram had only 4. What did * mean, which two groups’ comparison in Figure 2E-F, Figure 3A-D, and so on? More details are needed.

3) From the WB image in Figure 6B, there was no difference between d7 and d14 in UUO or Tlr2-/- group, which was different from the histogram result, and not in line with UUO progression. Besides, in the sham group, the expression of Galectin-3 on d14 was significantly reduced when compared to d7, why the expression of Galectin-3 in the sham group reduced.

4) In lines 78-80, the author described the protective effect of Tlr2-/- on inflammation and tubular injury. But the experiment results showed that Tlr2-/- had no influence on tubular inflammation, injury, or fibrosis. If it is paradox?

5) There was too much result explanation in the discussion of the manuscript. The experiment results showed TLR2 participated in tubular epithelial apoptosis, but had no influence on renal injury or fibrosis, neither proliferation nor inflammation. So what’s the role of TLR2 in UUO induced renal injury or fibrosis? More information about TLR2 on UUO model are needed.

6. PLOS authors have the option to publish the peer review history of their article (what does this mean?). If published, this will include your full peer review and any attached files.

Reviewer #1: No

Reviewer #2: No

---

## [Author Response · Author response to Decision Letter 0]

30 Jun 2023

Reviewer 1

1) The subject of this manuscript is TLR2 mediates renal apoptosis in neonatal mice subjected experimentally to obstructive nephropathy. However, in this manuscript, the proportion of narrative and experimental evidence of renal apoptosis is low, while other results that do not differ are slightly more discussed, indicating a shift in emphasis in the article. Authors could increase the types of experiments on renal apoptosis or dig deeper into the mechanism of TLR2 on renal apoptosis to make full-text logic more clarity.

Ad 1) Thank you for this important comment. We extended the section about apoptosis in our manuscript giving this topic the spotlight. You can find the additional text in lines 423-430. It reads now: ‘TLR2, as a part of the innate immune system is activated by bacterial lipoproteins and signals for apoptosis through MyD88 via a pathway involving Fas-associated death domain protein and caspase-8. Inflammation following UUO is not mediated by bacteria. UUO induces sterile inflammation, which is mediated by DAMPs, of which a variety might be able to activate the TLR2 mediated apoptotic pathway. High levels of apoptosis are associated with nephron loss in the developing kidney and thus the loss of renal function.’

We also reduced the sections in the discussion part regarding the remaining results.

2) Result I and II are both results about the center of the article, which is renal apoptosis, but they have somewhat smaller sample sizes, even less than the sample size of the undifferentiated results. It is recommended that researchers increase the sample size to make the article data more convincing.

Ad 2) The sample sizes for immunohistochemical staining are n=8 per group for every used staining method, in the case of apoptosis it’s the TUNEL staining. As for the western blot analysis, being here the analysis of caspase-8, we were only able to use 3 samples per analysis per group. Usually with adult mice it is possible to split the kidneys in parts for different analyses. However, we work with neonatal mice, of which kidney weights range on average between 15 mg at day 3 of life and only 65 mg on day 14 (sham-operated controls). The body weight of these neonatal mice also lies only between around 1.7 g on the second day of life, 4 g on day 7, and 7.3 g on day 14 of life [Lange-Sperandio B et al. Leukocytes Induce Epithelial to Mesenchymal Transition after Unilateral Ureteral Obstruction in Neonatal Mice. Am. J. Clin. Pathol. 2007; 861-871]. We have to perform a lysis of the whole kidney for enough material for our analyses. To ensure a large analytic range we are bound to limit our sample size used for protein analysis. 

3) The method section of the article does not describe the source of the knockout mice, and it is suggested to add a description to increase the credibility of the data. Moreover, the references of the article are old, it is recommended to use the journal literature within the last 3 years or 5 years if possible.

Ad 3) Thank you very much for your valuable suggestions. We added the source of the knockout mice to our manuscript, see lines 102-104. It reads now: ‘The Tlr2-/- mouse strain used (and crossed with other Tlr ko strains) has been generated by Deltagen, Cal, USA, and provided to CK through Tularik (merged into Amgen in the aftermath).’

We also updated the literature accordingly.

Reviewer 2

1) In Figure 1, the expression of TLR2 on d14 was less than d7 in the sham group, and in the UUO group the TLR2 expression on d7 and d14 was similar. How to explain this result?

Thank you for your comments. 

Ad 1) Thank you for this important comment. We work here with neonatal mice, in which nephrogenesis is still ongoing after birth. The changes in the TLR2 expression may be developmentally differently regulated. Studies have shown that the expression of TLR2 during organ development changes with time and depends heavily on the organ [Harju K, Glumoff V, Hallman M. Ontogeny of Toll-like receptors Tlr2 and Tlr4 in mice. Pediatr Res. 2001;49(1):81-3.]. If we than combine the results of the sham-operated and UUO kidneys, we see that the increase of TLR2 following UUO on d14 is indeed higher than on d7, which is consistent with the severity of the injury. 

2) In figure 2F and 3, there were 8 groups, but histogram had only 4. What did * mean, which two groups’ comparison in Figure 2E-F, Figure 3A-D, and so on? More details are needed.

Ad 2) Thank you very much for the question. We show the results of our analyses as x-fold change above sham-operated controls. This form of presentation allows us to show the actual impact of UUO with the sham-measurements as the basis. We divide the UUO measurements by the average of the sham measurements to achieve the results we present at the end. This way we reduce the presented groups to 4 (WT vs TLR2 x-fold above sham for d7 and d14). The * displayed alone here indicates that the differences between sham and UUO are significant. We updated the figure legends for more clarification, see lines 263-264, 290-291, 316-317, 367, 376-377. It reads now: ‘Standalone * represents significant differences between Sham and UUO results.’

3) From the WB image in Figure 6B, there was no difference between d7 and d14 in UUO or Tlr2-/- group, which was different from the histogram result, and not in line with UUO progression. Besides, in the sham group, the expression of Galectin-3 on d14 was significantly reduced when compared to d7, why the expression of Galectin-3 in the sham group reduced.

Ad 3) The differences we show here are x-fold above sham, so the decrease of α-SMA in the sham-operated controls impacts the shown results. The increase of α-SMA following UUO based on the sham results on d14 is higher than the increase on d7. This decrease of α-SMA has been shown before [Gasparitsch M et al. RAGE-mediated interstitial fibrosis in neonatal obstructive nephropathy is independent of NF-kappaB activation. Kidney Int. 2013;84(5):911-9.] in the sham-operated kidneys and is part of the ongoing nephrogenesis with mesenchymal-to-epithelial transition (MET). MET is a biological process that involves the transition from mesenchymal cells to epithelial cells as part of normal development. The drastic decrease observed in the expression of Galectin-3 is an effect of differently regulated proteins in the course of the development of the kidney. Galectin-3 plays a role in nephrogenesis [Winyard PJ et al. Epithelial galectin-3 during human nephrogenesis and childhood cystic diseases. J Am Soc Nephrol. 1997;8(11):1647-57.]. In the metanephros, the adult kidney precursor, galectin-3 was detected in the apical domains of ureteric bud branches, and there was intense expression in fetal medullary and papillary collecting ducts in both the cytoplasm and plasma membranes. Developmentally important proteins are expressed highly during nephrogenesis and decline with time, as the kidney development is gradually completed.

4) In lines 78-80, the author described the protective effect of Tlr2-/- on inflammation and tubular injury. But the experiment results showed that Tlr2-/- had no influence on tubular inflammation, injury, or fibrosis. If it is paradox?

Ad 4) We appreciate the comment of the referee. The protective effect of Tlr2-/- was shown in adult mice. However, we conducted our research on neonatal mice. We were able to show previously that neonatal mice can respond differently to UUO injury in comparison to adult mice [Kubik MJ et al. Renal developmental genes are differentially regulated after unilateral ureteral obstruction in neonatal and adult mice. Sci Rep. 2020;10(1):19302.]. We added „adult“ to this text part for further clarification.

5) There was too much result explanation in the discussion of the manuscript. The experiment results showed TLR2 participated in tubular epithelial apoptosis, but had no influence on renal injury or fibrosis, neither proliferation nor inflammation. So what’s the role of TLR2 in UUO induced renal injury or fibrosis? More information about TLR2 on UUO model are needed.

Ad 5) Thank you for pointing this out. We reduced the discussion text as suggested. TLR2 seems to play a minor role in neonatal UUO regarding renal injury, fibrosis, and inflammation. It is possible that alternative or additional signaling pathways are activated in neonatal mice following UUO and the knockout of TLR2 is not effective in ameliorating the whole injury, but just apoptosis. Apoptosis itself is a non-inflammatory cell death and its inhibition does not influence inflammation [Roychowdhury S, Chiang DJ, Mandal P, McMullen MR, Liu X, Cohen JI, et al. Inhibition of apoptosis protects mice from ethanol-mediated acceleration of early markers of CCl4-induced fibrosis but not steatosis or inflammation. Alcohol Clin Exp Res. 2012;36(7):1139-47]. The inhibition of other pathways in addition of TLR2 could give promising results and help to reduce the injury done to the kidneys until obstruction is relieved.

---

## [Decision Letter · Decision Letter 1]

12 Sep 2023

PONE-D-23-12099R1TLR2 mediates renal apoptosis in neonatal mice subjected experimentally to obstructive nephropathyPLOS ONE

Dear Dr. Lange-Sperandio,

Thank you for submitting your manuscript to PLOS ONE. After careful consideration, we feel that it has merit but does not fully meet PLOS ONE’s publication criteria as it currently stands. Therefore, we invite you to submit a revised version of the manuscript that addresses the points raised during the review process.

One reviewer still has raised some question, which I would like you to answer before acceptance.

We look forward to receiving your revised manuscript.

Kind regards,

Franziska Theilig

Academic Editor

PLOS ONE

Journal Requirements:

Reviewers' comments:

Reviewer's Responses to Questions

**Comments to the Author**

1. If the authors have adequately addressed your comments raised in a previous round of review and you feel that this manuscript is now acceptable for publication, you may indicate that here to bypass the “Comments to the Author” section, enter your conflict of interest statement in the “Confidential to Editor” section, and submit your "Accept" recommendation.

Reviewer #1: (No Response)

2. Is the manuscript technically sound, and do the data support the conclusions?

Reviewer #1: Partly

3. Has the statistical analysis been performed appropriately and rigorously? 

Reviewer #1: Yes

4. Have the authors made all data underlying the findings in their manuscript fully available?

Reviewer #1: Yes

5. Is the manuscript presented in an intelligible fashion and written in standard English?

Reviewer #1: Yes

6. Review Comments to the Author

Reviewer #1: Thank you to the authors for answering all my questions carefully and patiently.

The authors have added the section in the center of the article and shortened the discussion of the remaining sections accordingly, and I fully understand the explanation of the sample size; there is indeed a huge difference between neonatal and adult mice, and it is also a great thing to allocate the experimental materials wisely. Many thanks for adding the source of the knockout rats, which makes the source of the article's data even more compelling, but there are a few other minor problems with the article：

1、 In result 1“Neonatal UUO induces protein expression of TLR2”,authors only showed the results of Western Blotting experiment, I suggest that authors can add the pathological results of the sham group versus the UUO group,moreover,authors are able to testify the indicators of renal fibrosis.

2、 In result 2“TLR2 mediates tubular and interstitial apoptosis in neonatal kidneys with UUO”;Although authors used the knockout mouse,did not demonstrate the evidences of decreasing the expression of TlR2.On the other hand,the indicators of Western Blotting verifying apoptosis are insufficient,authors can add Bax and Bcl-2 etc.

3、 The results of subsequent experiments by the author were all negative,however,displaying experiments were inadequate to demonstrate the regulation of TLR2 on renal apoptosis.I recommend authors to supplement experiments to make the paper more logical and compact.

7. PLOS authors have the option to publish the peer review history of their article (what does this mean?). If published, this will include your full peer review and any attached files.

Reviewer #1: No

---

## [Author Response · Author response to Decision Letter 1]

19 Oct 2023

Response to Reviewers

Reviewer 1

1) In result 1“Neonatal UUO induces protein expression of TLR2”, authors only showed the results of Western Blotting experiment, I suggest that authors can add the pathological results of the sham group versus the UUO group, moreover, authors are able to testify the indicators of renal fibrosis.

Ad 1) Thank you for this valuable comment. We absolutely agree that it is important to show the pathological results of UUO. The pathological effects of neonatal UUO are thoroughly studied [1]. We indicate these changes in our study as well. The standalone asterisks (*) in the bar plots indicate that the results shown were significantly different between the sham and UUO group. This way we are able to show the overall pathology of neonatal UUO and the differences between Tlr2-/- and WT mice in one plot.

2) In result 2 “TLR2 mediates tubular and interstitial apoptosis in neonatal kidneys with UUO”; Although authors used the knockout mouse, did not demonstrate the evidence of decreasing the expression of TLR2. On the other hand, the indicators of Western Blotting verifying apoptosis are insufficient, authors can add Bax and Bcl-2 etc.

Ad 2) Thank you for this important suggestion. We agree that evidence that the knock-out mice indeed do not express TLR2 is essential. We added to figure 1 the results of a western blot analysis showing, that in comparison to WT the Tlr2-/- mice do not express TLR2 protein. We also added a text to the results section regarding this analysis, you can find it in lines 224-225 and it reads: “This analysis was also used to confirm that Tlr2-/- mice indeed did not express TLR2 (Fig 1B).”

Thank you for the comment regarding additional apoptosis experiments. We performed additional western blots of Bcl-2 and Bax. The protein expression of Bcl2 after ureteral obstruction increased significantly in Tlr2-/- mice at day 14 of life in comparison to WT. As Bcl-2 is known to suppress apoptosis [2], this confirms our findings, that TLR2 mediates apoptosis in the UUO model and that Tlr2-/- kidneys show less apoptosis after neonatal UUO in comparison to WT. We also measured the protein expression of Bax, an apoptosis activator [2] in neonatal mice kidneys. There was no significant difference in Bax expression between Tlr2-/- and WT mice. We added the results of both analyses to figure 3 and included these in the results and discussion sections in lines 259-264 and 441-443. It reads now: “For further analysis of cell death in our model we analyzed the anti-apoptotic marker Bcl-2 and the pro-apoptotic marker Bax using western blot (Fig 3). Neonatal Tlr2-/- mice showed a higher expression of Bcl-2 at day 14 in comparison to WT (Fig 3A), confirming that TLR2 mediates apoptosis in the neonatal model of obstructive nephropathy. Bax expression increased following UUO at day 14, without significant differences between Tlr2-/- and WT kidneys.” And (discussion) “As our results were conflicting, we decided to include Bcl-2 as an anti-apoptotic marker [2, 3]. The increase of Bcl-2 expression in Tlr2-/- kidneys further confirms our findings that TLR2 mediates apoptosis in neonatal UUO.”.

3) The results of subsequent experiments by the author were all negative, however, displaying experiments were inadequate to demonstrate the regulation of TLR2 on renal apoptosis. I recommend authors to supplement experiments to make the paper more logical and compact.

Ad 3) Thank you for this valuable insight. We agree that the displayed results were not strong enough to show that TLR2 mediates apoptosis in the neonatal UUO model. We hope that the additional experiments we performed are sufficient to confirm our findings in this study. We think that these results have important implications for future research on TLR2.

1. Chevalier, R.L., M.S. Forbes, and B.A. Thornhill, Ureteral obstruction as a model of renal interstitial fibrosis and obstructive nephropathy. Kidney Int, 2009. 75(11): p. 1145-1152.

2. Chipuk, J.E., et al., The BCL-2 family reunion. Mol Cell, 2010. 37(3): p. 299-310.

3. Zhang, G., et al., Role of apoptosis and Bcl-2/Bax in the development of tubulointerstitial fibrosis during experimental obstructive nephropathy. Exp Nephrol, 2001. 9(2): p. 71-80.

---

## [Editor Report · Decision Letter 2]

26 Oct 2023

TLR2 mediates renal apoptosis in neonatal mice subjected experimentally to obstructive nephropathy

PONE-D-23-12099R2

Dear Dr. Lange-Sperandio,

We’re pleased to inform you that your manuscript has been judged scientifically suitable for publication and will be formally accepted for publication once it meets all outstanding technical requirements.

Kind regards,

Franziska Theilig

Academic Editor

PLOS ONE
---

## [Editor Report · Acceptance letter]

16 Nov 2023

PONE-D-23-12099R2 

TLR2 mediates renal apoptosis in neonatal mice subjected experimentally to obstructive nephropathy 

Dear Dr. Lange-Sperandio:

I'm pleased to inform you that your manuscript has been deemed suitable for publication in PLOS ONE. Congratulations! Your manuscript is now with our production department. 

Kind regards, 

on behalf of

Dr. Franziska Theilig 

Academic Editor

PLOS ONE